# KERNEL CHANGE-POINT DETECTION WITH AUXILIARY DEEP GENERATIVE MODELS

**Wei-Cheng Chang, Chun-Liang Li, Yiming Yang & Barnabás Póczos**
Carnegie Mellon University
Pittsburgh, PA 15213, USA
{wchang2,chunlial,yiming,bapoczos}@cs.cmu.edu

## ABSTRACT

Detecting the emergence of abrupt property changes in time series is a challenging problem. Kernel two-sample test has been studied for this task which makes fewer assumptions on the distributions than traditional parametric approaches. However, selecting kernels is non-trivial in practice. Although kernel selection for two-sample test has been studied, the insufficient samples in change point detection problem hinders the success of those developed kernel selection algorithms. In this paper, we propose **KL-CPD**, a novel kernel learning framework for time series CPD that optimizes a lower bound of test power via an auxiliary generative model. With deep kernel parameterization, **KL-CPD** endows kernel two-sample test with the data-driven kernel to detect different types of change-points in real-world applications. The proposed approach significantly outperformed other state-of-the-art methods in our comparative evaluation of benchmark datasets and simulation studies.

## 1 INTRODUCTION

Detecting changes in the temporal evolution of a system (biological, physical, mechanical, etc.) in time series analysis has attracted considerable attention in machine learning and data mining for decades (Basseville et al., 1993; Brodsky & Darkhovsky, 2013). This task, commonly referred to as change-point detection (CPD) or anomaly detection in the literature, aims to predict significant changing points in a temporal sequence of observations. CPD has a broad range of real-world applications such as medical diagnostics (Gardner et al., 2006), industrial quality control (Basu & Meckesheimer, 2007), financial market analysis (Pepelyshev & Polunchenko, 2015), video anomaly detection (Liu et al., 2018) and more.

As shown in Fig. 1, we focus on the retrospective CPD (Takeuchi & Yamanishi, 2006; Li et al., 2015a), which allows a flexible time window to react on the change-points. Retrospective CPD not only enjoys more robust detection (Chandola et al., 2009) but embraces many applications such as climate change detection (Reeves et al., 2007), genetic sequence analysis (Wang et al., 2011), networks intrusion detection (Yamanishi et al., 2004), to name just a few. Various methods have been developed (Gustafsson & Gustafsson, 2000), and many of them are parametric with strong assumptions on the distributions (Basseville et al., 1993; Gustafsson, 1996), including auto-regressive models (Yamanishi & Takeuchi, 2002) and state-space models (Kawahara et al., 2007) for tracking changes in the mean, the variance, and the spectrum.

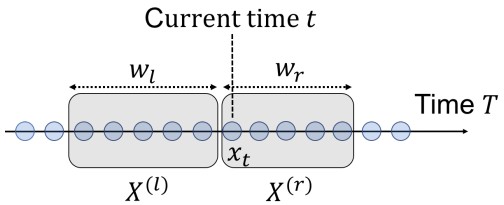

Figure 1: A sliding window over the time series input with two intervals: the past and the current, where $w_l, w_r$ are the size of the past and current interval, respectively. $X^{(l)}, X^{(r)}$ consists of the data in the past and current interval, respectively.

Ideally, the detection algorithm should be free of distributional assumptions to have robust performance as neither true data distributions nor anomaly types are known a priori. Thus the parametric

assumptions in many works are unavoidably a limiting factor in practice. As an alternative, nonparametric and kernel approaches are free of distributional assumptions and hence enjoy the advantage to produce more robust performance over a broader class of data distributions.

Kernel two-sample test has been applied to time series CPD with some success. For example, Harchaoui et al. (2009) presented a test statistic based upon the maximum kernel fisher discriminant ratio for hypothesis testing and Li et al. (2015a) proposed a computational efficient test statistic based on maximum mean discrepancy with block sampling techniques. The performance of kernel methods, nevertheless, relies heavily on the choice of kernels. Gretton et al. (2007; 2012a) conducted kernel selection for RBF kernel bandwidths via median heuristic. While this is certainly straightforward, it has no guarantees of optimality regarding to the statistical test power of hypothesis testing. Gretton et al. (2012b) show explicitly optimizing the test power leads to better kernel choice for hypothesis testing under mild conditions. Kernel selection by optimizing the test power, however, is not directly applicable for time series CPD due to insufficient samples, as we discuss in Section 3.

In this paper, we propose **KL-CPD**, a kernel learning framework for time series CPD. Our main contributions are three folds.

- In Section 3, we first observe the inaptness of existing kernel learning approaches in a simulated example. We then propose to optimize a lower bound of the test power via an auxiliary generative model, which aims at serving as a surrogate of the abnormal events.

- In Section 4, we present a deep kernel parametrization of our framework, which endows a data-driven kernel for the kernel two-sample test. **KL-CPD** induces composition kernels by combining RNNs and RBF kernels that are suitable for the time series applications.

- In Section 5, we conduct extensive benchmark evaluation showing the outstanding performance of **KL-CPD** in real-world CPD applications. With simulation-based analysis in Section 6, in addition, we can see the proposed method not only boosts the kernel power but also evades the performance degradation as data dimensionality of time series increases.

Finally, our experiment code and datasets are available at https://github.com/OctoberChang/klcpd_code.

## 2 PRELIMINARY

Given a sequence of $d$-dimensional observations $\{x_1, \ldots, x_t, \ldots\}, x_i \in \mathbb{R}^d$, our goal is to detect the existence of a change-point[1] such that before the change-point, samples are i.i.d from a distribution $\mathbb{P}$, while after the change-point, samples are i.i.d from a different distribution $\mathbb{Q}$. Suppose at current time $t$ and the window size $w$, denote the past window segment $X^{(l)} = \{x_{t-w}, \ldots, x_{t-1}\}$ and the current window segment $X^{(r)} = \{x_t, \ldots, x_{t+w-1}\}$, We compute the maximum mean discrepancy (MMD) between $X^{(l)}$ and $X^{(r)}$, and use it as the plausibility of change-points: The higher the distribution discrepancy, the more likely the point is a change-point.

Notice that there are multiple settings for change point detection (CPD) where samples could be piecewise iid, non-iid autoregressive, and more. It is truly difficult to come up with a generic framework to tackle all these different settings. In this paper, following the previous CPD works (Harchaoui et al., 2009; Kawahara et al., 2007; Matteson & James, 2014; Li et al., 2015a), we stay with the piecewise iid assumption of the time series samples. Extending the current model to other settings is interesting and we leave it for future work.

### 2.1 MMD AND TEST POWER

We review maximum mean discrepancy (MMD) and its use to two-sample test, which are two cornerstones in this work. Let $k$ be the kernel of a reproducing kernel Hilbert space (RKHS) $\mathcal{H}_k$ of functions on a set $\mathcal{X}$. We assume that $k$ is measurable and bounded, $\sup_{x \in \mathcal{X}} k(x, x) < \infty$. MMD

---

[1]Technically speaking, two-sample test only informs whether two finite sample sets are from the same distribution or not. We abuse the change-point notation as equivalent to that two sample sets differs in distribution. One can further apply other segmentation techniques such as maximum partition strategy to locate a single change point after the detection of two-sample test.

is a nonparametric probabilistic distance commonly used in two-sample-test (Gretton et al., 2007; 2012a). Given a kernel $k$, the MMD distance between two distributions $\mathbb{P}$ and $\mathbb{Q}$ is defined as

$$M_k(\mathbb{P}, \mathbb{Q}) := \|\mu_\mathbb{P} - \mu_\mathbb{Q}\|_{\mathcal{H}_k}^2 = \mathbb{E}_\mathbb{P}[k(x, x')] - 2\mathbb{E}_{\mathbb{P}, \mathbb{Q}}[k(x, y)] + \mathbb{E}_\mathbb{Q}[k(y, y')],$$

where $\mu_\mathbb{P} = \mathbb{E}_{x \sim \mathbb{P}}[k(x, \cdot)], \mu_\mathbb{Q} = \mathbb{E}_{y \sim \mathbb{Q}}[k(y, \cdot)]$ are the kernel mean embedding for $\mathbb{P}$ and $\mathbb{Q}$, respectively. In practice we use finite samples from distributions to estimate MMD distance. Given $X = \{x_1, \ldots, x_m\} \sim \mathbb{P}$ and $Y = \{y_1, \ldots, y_m\} \sim \mathbb{Q}$, one unbiased estimator of $M_k(\mathbb{P}, \mathbb{Q})$ is

$$\hat{M}_k(X, Y) := \frac{1}{\binom{m}{2}} \sum_{i \neq i'} k(x_i, x_{i'}) - \frac{2}{m^2} \sum_{i,j} k(x_i, y_j) + \frac{1}{\binom{m}{2}} \sum_{j \neq j'} k(y_j, y_{j'}).$$

which has nearly minimal variance among unbiased estimators (Gretton et al., 2012a, Lemma 6).

For any characteristic kernel $k$, $M_k(\mathbb{P}, \mathbb{Q})$ is non-negative and in particular $M_k(\mathbb{P}, \mathbb{Q}) = 0$ *iff* $\mathbb{P} = \mathbb{Q}$. However, the estimator $\hat{M}_k(X, X')$ may not be 0 even though $X, X' \sim \mathbb{P}$ due to finite sample size. Hypothesis test instead offers thorough statistical guarantees of whether two finite sample sets are the same distribution. Following Gretton et al. (2012a), the hypothesis test is defined by the null hypothesis $H_0 : \mathbb{P} = \mathbb{Q}$ and alternative $H_1 : \mathbb{P} \neq \mathbb{Q}$, using test statistic $m\hat{M}_k(X, Y)$. For a given allowable false rejection probability $\alpha$ (i.e., false positive rate or Type I error), we choose a test threshold $c_\alpha$ and reject $H_0$ if $m\hat{M}_k(X, Y) > c_\alpha$.

We now describe the objective to choose the kernel $k$ for maximizing the test power (Gretton et al., 2012b; Sutherland et al., 2017). First, note that, under the alternative $H_1 : \mathbb{P} \neq \mathbb{Q}$, $\hat{M}_k$ is asymptotically normal,

$$\frac{\hat{M}_k(X, Y) - M_k(\mathbb{P}, \mathbb{Q})}{\sqrt{V_m(\mathbb{P}, \mathbb{Q})}} \xrightarrow{\mathcal{D}} \mathcal{N}(0, 1), \tag{1}$$

where $V_m(\mathbb{P}, \mathbb{Q})$ denotes the asymptotic variance of the $\hat{M}_k$ estimator. The test power is then

$$\Pr\left(m\hat{M}_k(X, Y) > c_\alpha\right) \longrightarrow \Phi\left(\frac{M_k(\mathbb{P}, \mathbb{Q})}{\sqrt{V_m(\mathbb{P}, \mathbb{Q})}} - \frac{c_\alpha}{m\sqrt{V_m(\mathbb{P}, \mathbb{Q})}}\right) \tag{2}$$

where $\Phi$ is the CDF of the standard normal distribution. Given a set of kernels $\mathcal{K}$, We aim to choose a kernel $k \in \mathcal{K}$ to maximize the test power, which is equivalent to maximizing the argument of $\Phi$.

## 3 OPTIMIZING TEST POWER FOR CHANGE-POINT DETECTION

In time series CPD, we denote $\mathbb{P}$ as the distribution of usual events and $\mathbb{Q}$ as the distribution for the event when change-points happen. The difficulty of choosing kernels via optimizing test power in Eq. (2) is that we have very limited samples from the abnormal distribution $\mathbb{Q}$. Kernel learning in this case may easily overfit, leading to sub-optimal performance in time series CPD.

### 3.1 DIFFICULTIES OF OPTIMIZING KERNELS FOR CPD

To demonstrate how limited samples of $\mathbb{Q}$ would affect optimizing test power, we consider kernel selection for Gaussian RBF kernels on the Blobs dataset (Gretton et al., 2012b; Sutherland et al., 2017), which is considered hard for kernel two-sample test. $\mathbb{P}$ is a $5 \times 5$ grid of two-dimensional standard normals, with spacing 15 between the centers. $\mathbb{Q}$ is laid out identically, but with covariance $\frac{\epsilon_q - 1}{\epsilon_q + 1}$ between the coordinates (so the ratio of eigenvalues in the variance is $\epsilon_q$). Left panel of Fig. 2 shows $X \sim \mathbb{P}$ (red samples), $Y \sim \mathbb{Q}$ (blue dense samples), $\tilde{Y} \sim \mathbb{Q}$ (blue sparse samples) with $\epsilon_q = 6$. Note that when $\epsilon_q = 1$, $\mathbb{P} = \mathbb{Q}$.

For $\epsilon_q \in \{4, 6, 8, 10, 12, 14\}$, we take 10000 samples for $X, Y$ and 200 samples for $\tilde{Y}$. We consider two objectives for choosing kernels: 1) *median heuristic*; 2) *max-ratio* $\eta_{k^*}(X, Y) = \arg\max_k \hat{M}_k(X, Y)/\sqrt{V_m(X, Y)}$; among 20 kernel bandwidths. We repeat this process 1000 times and report the test power under false rejection rate $\alpha = 0.05$. As shown in the right panel of Fig. 2, optimizing kernels using limited samples $\tilde{Y}$ significantly decreases the test power compared to $Y$ (blue curve down to the cyan curve). This result not only verifies our claim on the inaptness of existing kernel learning objectives for CPD task, but also stimulates us with the following question, *How to optimize kernels with very limited samples from $\mathbb{Q}$, even none in an extreme?*

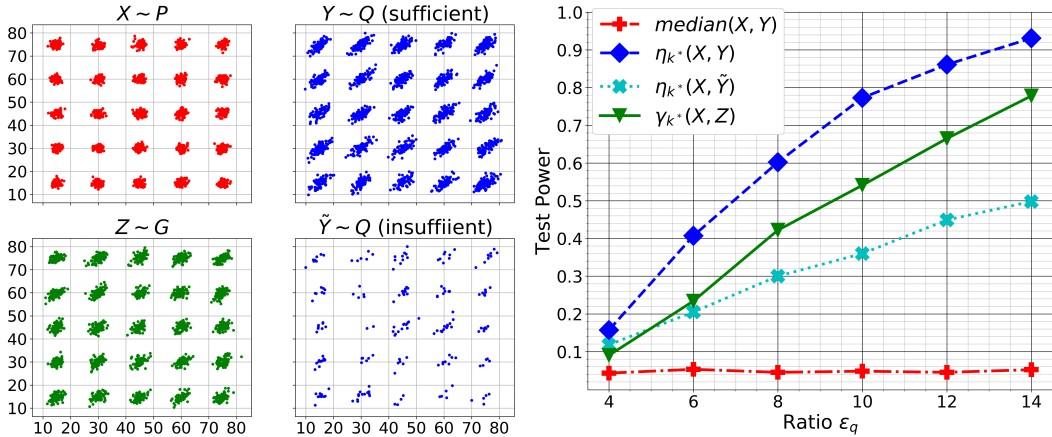

Figure 2: **Left**: $5 \times 5$ Gaussian grid, samples from $\mathbb{P}$, $\mathbb{Q}$ and $\mathbb{G}$. We discuss two cases of $\mathbb{Q}$, one of sufficient samples, the other of insufficient samples. **Right**: Test power of kernel selection versus $\epsilon_q$. Choosing kernels by $\gamma_{k^*}(X, Z)$ using a surrogate distribution $\mathbb{G}$ is advantageous when we do not have sufficient samples from $\mathbb{Q}$, which is typically the case in time series CPD task.

## 3.2 A PRACTICAL LOWER BOUND ON OPTIMIZING TEST POWER

We first assume there exist a surrogate distribution $\mathbb{G}$ that we can easily draw samples from ($Z \sim \mathbb{G}$, $|Z| \gg |\tilde{Y}|$), and also satisfies the following property:

$$M_k(\mathbb{P}, \mathbb{P}) < M_k(\mathbb{P}, \mathbb{G}) < M_k(\mathbb{P}, \mathbb{Q}), \forall k \in \mathcal{K}, \tag{3}$$

Besides, we assume dealing with non trivial case of $\mathbb{P}$ and $\mathbb{Q}$ where a lower bound $\frac{1}{m} v_l \leq V_{m,k}(\mathbb{P}, \mathbb{Q}), \forall k$ exists. Since $M_k(\mathbb{P}, \mathbb{Q})$ is bounded, there exists an upper bound $v_u$. With bounded variance $\frac{v_l}{m} \leq V_{m,k}(\mathbb{P}, \mathbb{Q}) \leq \frac{v_u}{m}$ condition, we derive an lower bound $\gamma_{k^*}(\mathbb{P}, \mathbb{G})$ of the test power

$$\max_{k \in \mathcal{K}} \frac{M_k(\mathbb{P}, \mathbb{Q})}{\sqrt{V_m(\mathbb{P}, \mathbb{Q})}} - \frac{c_\alpha/m}{\sqrt{V_m(\mathbb{P}, \mathbb{Q})}} \geq \max_{k \in \mathcal{K}} \frac{M_k(\mathbb{P}, \mathbb{Q})}{\sqrt{v_u/m}} - \frac{c_\alpha}{\sqrt{mv_l}} \geq \max_{k \in \mathcal{K}} \frac{M_k(\mathbb{P}, \mathbb{G})}{\sqrt{v_u/m}} - \frac{c_\alpha}{\sqrt{mv_l}} = \gamma_{k^*}(\mathbb{P}, \mathbb{G}).$$
$$\tag{4}$$

Just for now in the blob toy experiment, we artifact this distribution $\mathbb{G}$ by mimicking $\mathbb{Q}$ with the covariance $\epsilon_g = \epsilon_q - 2$. We defer the discussion on how to find $\mathbb{G}$ in the later subsection 3.3. Choosing kernels via $\gamma_{k^*}(X, Z)$ using surrogate samples $Z \sim \mathbb{G}$, as represented by the green curve in Fig. 2, substantially boosts the test power compared to $\eta_{k*}(X, \tilde{Y})$ with sparse samples $\tilde{Y} \sim \mathbb{Q}$. This toy example not only suggesets that optimizing kernel with surrogate distribution $\mathbb{G}$ leads to better test power when samples from $\mathbb{Q}$ are insufficient, but also demonstrates that the effectiveness of our kernel selection objective holds without introducing any autoregressive/RNN modeling to control the Type-I error.

**Test Threshold Approximation** Under $H_0 : \mathbb{P} = \mathbb{Q}$, $m\hat{M}_k(X, Y)$ converges asymptotically to a distribution that depends on the unknown data distribution $\mathbb{P}$ (Gretton et al., 2012a, Theorem 12); we thus cannot evaluate the test threshold $c_\alpha$ in closed form. Common ways of estimating threshold includes the permutation test and a estimated null distribution based on approximating the eigenspectrum of the kernel. Nonetheless, both are still computational demanding in practice. Even with the estimated threshold, it is difficult to optimize $c_\alpha$ because it is a function of $k$ and $\mathbb{P}$.

For $X, X' \sim \mathbb{P}$, we know that $c_\alpha$ is a function of the empirical estimator $\hat{M}_k(X, X')$ that controls the Type I error. Bounding $\hat{M}_k(X, X')$ could be an approximation of bounding $c_\alpha$. Therefore, we propose the following objective that maximizing a lower bound of test power

$$\underset{k \in \mathcal{K}}{\arg\max}\, M_k(\mathbb{P}, \mathbb{G}) - \lambda \hat{M}_k(X, X'), \tag{5}$$

where $\lambda$ is a hyper-parameter to control the trade-off between Type-I and Type-II errors, as well as absorbing the constants $m, v_l, v_u$ in variance approximation. Note that in experiment, the optimization of Eq. (5) is solved using the unbiased estimator of $M_k(\mathbb{P}, \mathbb{G})$ with empirical samples.

### 3.3 Surrogate Distributions using Generative Models

The remaining question is how to construct the surrogate distribution $\mathbb{G}$ without any sample from $\mathbb{Q}$. Injecting random noise to $\mathbb{P}$ is a simple way to construct $\mathbb{G}$. While straightforward, it may result in a sub-optimal $\mathbb{G}$ because of sensitivity to the level of injected random noise. As no prior knowledge of $\mathbb{Q}$, to ensure (3) hold for any possible $\mathbb{Q}$ (e.g. $\mathbb{Q} \neq \mathbb{P}$ but $\mathbb{Q} \approx \mathbb{P}$), intuitively, we have to make $\mathbb{G}$ as closed to $\mathbb{P}$ as possible. We propose to learn an *auxiliary generative model* $\mathbb{G}_\theta$ parameterized by $\theta$ such that

$$\hat{M}_k(X, X') < \min_\theta M_k(\mathbb{P}, \mathbb{G}_\theta) < M_k(\mathbb{P}, \mathbb{Q}), \forall k \in \mathcal{K}.$$

To ensure the first inequality hold, we set early stopping criterion when solving $\mathbb{G}_\theta$ in practice. Also, if $\mathbb{P}$ is sophisticate, which is common in time series cases, limited capacity of parametrization of $\mathbb{G}_\theta$ with finite size model (e.g. neural networks) (Arora et al., 2017) and finite samples of $\mathbb{P}$ also hinder us to fully recover $\mathbb{P}$. Therefore, we result in a min-max formulation to consider all possible $k \in \mathcal{K}$ when we learn $\mathbb{G}$,

$$\min_\theta \max_{k \in \mathcal{K}} \quad M_k(\mathbb{P}, \mathbb{G}_\theta) - \lambda \hat{M}_k(X, X'), \tag{6}$$

and solve the kernel for the hypothesis test in the mean time. In experiment, we use simple alternative (stochastic) gradient descent to solve each other.

Lastly, we remark that although the resulted objective (6) is similar to Li et al. (2017), *the motivation and explanation are different*. One major difference is we aim to find $k$ with highest test power while their goal is finding $\mathbb{G}_\theta$ to approximate $\mathbb{P}$. A more detailed discussion can be found in Appendix A.

## 4 KLCPD: Realization for Time Series Applications

In this section, we present a realization of the kernel learning framework for time series CPD.

**Compositional Kernels** To have a more expressive kernel for complex time series, we consider compositional kernels $\tilde{k} = k \circ f$ that combines RBF kernels $k$ with injective functions $f_\phi$:

$$K = \left\{ \tilde{k} \mid \tilde{k}(x, x') = \exp(-\|f_\phi(x) - f_\phi(x)'\|^2) \right\}. \tag{7}$$

The resulted kernel $\tilde{k}$ is still characteristic if $f$ is an injective function and $k$ is characteristic (Gretton et al., 2012a). This ensures the MMD endowed by $\tilde{k}$ is still a valid probabilistic distance. One example function class is $\{f_\phi | f_\phi(x) = \phi x, \phi > 0\}$, equivalent to the kernel bandwidth tuning. Inspired by the recent success of combining deep neural networks into kernels (Wilson et al., 2016; Al-Shedivat et al., 2017; Li et al., 2017), we parameterize the injective functions $f_\phi$ by recurrent neural networks (RNNs) to capture the temporal dynamics of time series.

For an injective function $f$, there exists a function $F$ such that $F(f(x)) = x, \forall x \in \mathcal{X}$, which can be approximated by an auto-encoder via sequence-to-sequence architecture for time series. One practical realization of $f$ would be a RNN encoder parametrized by $\phi$ while the function $F$ is a RNN decoder parametrized by $\psi$ trained to minimize the reconstruction loss. Thus, our final objective is

$$\min_\theta \max_\phi \quad M_{f_\phi}(\mathbb{P}, \mathbb{G}_\theta) - \lambda \cdot \hat{M}_{f_\phi}(X, X') - \beta \cdot \mathbb{E}_{\nu \in \mathbb{P} \cup \mathbb{G}_\theta} \|\nu - F_\psi(f_\phi(\nu))\|_2^2. \tag{8}$$

**Practical Implementation** In practice, we consider two consecutive windows in mini-batch to estimate $\hat{M}_{f_\phi}(X, X')$ in an online fashion for the sake of efficiency. Specifically, the sample $X \sim \mathbb{P}$ is divided into the left window segment $X^{(l)} = \{x_{t-w}, \ldots, x_{t-1}\}$ and the right window segment $X^{(r)} = \{x_t, \ldots, x_{t+w-1}\}$ such that $X = \{X^{(l)}, X^{(r)}\}$. We now reveal implementation details of the auxiliary generative model and the deep kernel.

**Generator** $g_\theta$ Instead of modeling the explicit density $\mathbb{G}_\theta$, we model a generator $g_\theta$ where we can draw samples from. The goal of $g_\theta$ is to generate plausibly counterfeit but natural samples based on historical $X \sim \mathbb{P}$, which is similar to the conditional GANs (Mirza & Osindero, 2014; Isola et al., 2017). We use sequence-to-sequence (Seq2Seq) architectures (Sutskever et al., 2014) where

$g_{\theta_e}$ encodes time series into hidden states, and $g_{\theta_d}$ decodes it with the distributional autoregressive process to approximate the surrogate sample $Z$:

$$H = g_{\theta_e}\big(X^{(l)}, \mathbf{0}\big), \quad \tilde{h} = h_{t-1} + \omega, \quad Z = g_{\theta_d}\big(X^{(r)}_{\gg 1}, \tilde{h}\big).$$

where $\omega \sim \mathbb{P}(W)$ is a $d_h$-dimensional random noise sampled from a base distribution $\mathbb{P}(W)$ (e.g., uniform, Gaussian). $H = [h_{t-w}, \ldots, h_{t-1}] \in \mathbb{R}^{d_h \times w}$ is a sequence of hidden states of the generator's encoder. $X^{(r)}_{\gg 1} = \{\mathbf{0}, x_t, x_{t+1}, \ldots, x_{t+w-2}\}$ denotes right shift one unit operator over $X^{(r)}$.

**Deep Kernel Parametrization** We aim to maximize a lower bound of test power via back-propagation on $\phi$ using the deep kernel form $\tilde{k} = k \circ f_\phi$. On the other hand, we can also view the deep kernel parametrization as an **embedding learning** on the injective function $f_\phi(x)$ that can be distinguished by MMD. Similar to the design of generator, the deep kernel is a Seq2Seq framework with one GRU layer of the follow form:

$$H_\nu = f_\phi\big(\nu\big), \ \hat{\nu} = F_\psi\big(H_\nu\big).$$

where $\nu \sim \mathbb{P} \cup \mathbb{G}_\theta$ are from either the time series data $X$ or the generated sample $Z \sim g_\theta(\omega|X)$.

We present an realization of **KL-CPD** in Algorithm 1 with the weight-clipping technique. The stopping condition is based on a maximum number of epochs or the detecting power of kernel MMD $M_{f_\phi}\big(\mathbb{P}, \mathbb{G}_\theta\big) \leq \epsilon$. This ensure the surrogate $\mathbb{G}_\theta$ is not too close to $\mathbb{P}$, as motivated in Sec. 3.2.

---

**Algorithm 1: KL-CPD**, our proposed algorithm.

---

**input** : $\alpha$ the learning rate, $c$ the clipping parameter, $w$ the window size, $n_c$ the number of iterations of deep kernels training per generator update.

**while** $M_{k \circ f_\phi}(\mathbb{P}, \mathbb{G}_\theta) > \epsilon$ **do**

    **for** $t = 1, \ldots, n_c$ **do**

        Sample a minibatch $X_t \sim \mathbb{P}$, denote $X_t = \{X_t^{(l)}, X_t^{(r)}\}$, and $\omega \sim \mathbb{P}(\Omega)$

        gradient$(\phi) \leftarrow \nabla_\phi M_{k \circ f_\phi}\big(\mathbb{P}, \mathbb{G}_\theta\big) - \lambda \hat{M}_{k \circ f_\phi}\big(X_t^{(l)}, X_t^{(r)}\big) - \beta \mathbb{E}_{\nu \sim \mathbb{P} \cup \mathbb{G}_\theta} \|\nu - F_\psi\big(f_\phi(\nu)\big)\|_2^2$

        $\phi \leftarrow \phi + \alpha \cdot \text{RMSProp}(\phi, \text{gradient}(\phi))$

        $\phi \leftarrow \text{clip}(\phi, -c, c)$

    Sample a minibatch $X_{t'} \sim \mathbb{P}$, denote $X_{t'} = \{X_{t'}^{(l)}, X_{t'}^{(r)}\}$, and $\omega \sim \mathbb{P}(\Omega)$

    gradient$(\theta) \leftarrow \nabla_\theta M_{k \circ f_\phi}\big(\mathbb{P}, \mathbb{G}_\theta\big)$

    $\theta \leftarrow \theta - \alpha \cdot \text{Adam}(\theta, \text{gradient}(\theta))$

---

# 5 EVALUATION ON REAL-WORLD DATA

The section presents a comparative evaluation of the proposed **KL-CPD** and seven representative baselines on benchmark datasets from real-world applications of CPD, including the domains of biology, environmental science, human activity sensing, and network traffic loads. The data statistics are summarized in Table 1. We pre-process all dataset by normalizing each dimension in the range of $[0, 1]$. Detailed descriptions are available in Appendix B.1.

Following Lai et al. (2018); Saatçi et al. (2010); Liu et al. (2013), the datasets are split into the training set (60%), validation set (20%) and test set (20%) in chronological order. Note that training is fully unsupervised for all methods while labels in the validation set are used for hyperparameters tuning.

For quantitative evaluation, we consider re-ceiver operating characteristic (ROC) curves of

| Dataset | T | #sequences | domain | #labels |
|---|---|---|---|---|
| **Bee-Dance** | 826.66 | 6 | $\mathbb{R}^3$ | 19.5 |
| **Fishkiller** | 45175 | 1 | $\mathbb{R}^+$ | 899 |
| **HASC** | 39397 | 1 | $\mathbb{R}^3$ | 65 |
| **Yahoo** | 1432.13 | 15 | $\mathbb{R}^+$ | 36.06 |

Table 1: Dataset. $T$ is length of time series, #labels is average number of labeled change points.

anomaly detection results, and measure the area-under-the-curve (AUC) as the evaluation metric. AUC is commonly used in CPD literature (Li et al., 2015a; Liu et al., 2013; Xu et al., 2017).

We compare **KL-CPD** with real-time CPD methods (**ARMA**, **ARGP**, **RNN,LSTNet**) and retrospective CPD methods (**ARGP-BOCPD**, **RDR-KCPD**, **Mstats-KCPD**). Details are in Appendix B.3. Note that **OPT-MMD** is a deep kernel learning baseline which optimizes MMD by treating past samples as $\mathbb{P}$ and the current window as $\mathbb{Q}$ (insufficient samples).

| Method | Bee-Dance | Fishkiller | HASC | Yahoo |
|---|---|---|---|---|
| **ARMA** (Box, 2013) | 0.5368 | 0.8794 | 0.5863 | 0.8615 |
| **ARGP** (Candela et al., 2003) | 0.5833 | 0.8813 | 0.6448 | **0.9318** |
| **RNN** (Cho et al., 2014) | 0.5827 | 0.8872 | 0.6128 | 0.8508 |
| **LSTNet** (Lai et al., 2018) | 0.6168 | 0.9127 | 0.5077 | 0.8863 |
| **ARGP-BOCPD** (Saatçi et al., 2010) | 0.5089 | 0.8333 | 0.6421 | 0.9130 |
| **RDR-KCPD** (Liu et al., 2013) | 0.5197 | 0.4942 | 0.4217 | 0.6029 |
| **Mstats-KCPD** (Li et al., 2015a) | 0.5616 | 0.6392 | 0.5199 | 0.6961 |
| **OPT-MMD** | 0.5262 | 0.7517 | 0.6176 | 0.8193 |
| **KL-CPD** (Proposed method) | **0.6767** | **0.9596** | **0.6490** | 0.9146 |

Table 2: AUC on four real-world datasets. **KL-CPD** has the best AUC on three out of four datasets.

## 5.1 MAIN RESULTS

In Table 2, the first four rows present the real-time CPD methods, followed by three retrospective-CPD models, and the last is our proposed method. **KL-CPD** shows significant improvement over the other methods on all the datasets, except being in a second place on the **Yahoo** dataset, with 2% lower AUC compared to the leading **ARGP**. This confirms the importance of data-driven kernel selection and effectiveness of our kernel learning framework. Notice that **OPT-MMD** performs not so good compared to **KL-CPD**, which again verifies our simulated example in Sec. 3 that directly applying existing kernel learning approaches with insufficient samples may not be suitable for real-world CPD task.

Distribution matching approaches like **RDR-KCPD** and **Mstats-KCPD** are not as competitive as **KL-CPD**, and often inferior to real-time CPD methods. One explanation is both **RDR-KCPD** and **Mstats-KCPD** measure the distribution distance in the original data space with simple kernel selection using the median heuristic. The change-points may be hard to detect without the latent embedding learned by neural networks. **KL-CPD**, instead, leverages RNN to extract useful contexts and encodes time series in a discriminative embedding (latent space) on which kernel two-sample test is used to detection changing points. This also explains the inferior performance of **Mstats-KCPD** which uses kernel MMD with a fix RBF kernel. That is, using a fixed kernel to detect versatile types of change points is likely to fail.

Finally, the non-iid temporal structure in real-world applications may raise readers concern that the improvement coming from adopting RNN and controlling type-I error for model selection (kernel selection). Indeed, using RNN parameterized kernels (trained by minimizing reconstruction loss) buys us some gain compared to directly conduct kernel two-sample test on the original time series samples (Figure 3 cyan bar rises to blue bar). Nevertheless, we still have to do model selection to decide the parameters of RNN. In Table 2, we studied a kernel learning baseline, **OPT-MMD**, that optimizing an RNN parameterized kernel by controlling type-I error without the surrogate distribution. **OPT-MMD** is inferior to the **KL-CPD** that introduce the surrogate distribution with an auxiliary generator. On the other hand, from Table 2, we can also observe **KL-CPD** is better than other RNN alternatives, such as **LSTNet**. Those performance gaps between **KL-CPD**, **OPT-MMD** (regularizing type-I only) and other RNN works indicate the proposed maximizing testing power framework via an auxiliary distribution serves as a good surrogate for kernel (model) selection.

## 5.2 ABLATION TEST ON LEARNING KERNELS WITH DIFFERENT ENCODERS

We further examine how different encoders $f_\phi$ affects **KL-CPD**. For **MMD-dataspace**, $f_\phi$ is an identity map, equivalent to kernel selection with median heuristic in data space. For **MMD-codespace**, $\{f_\phi, F_\psi\}$ is a Seq2Seq autoencoder minimizing reconstruction loss without optimizing test power. For **MMD-negsample**, the same objective as **KL-CPD** except for replacing the auxiliary generator with injecting Gaussian noise to $\mathbb{P}$.

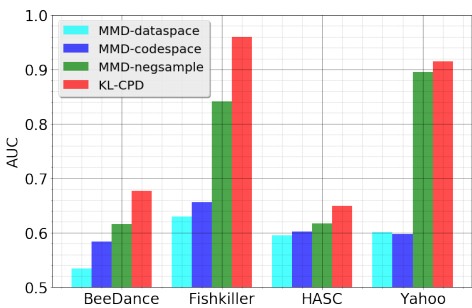
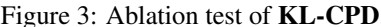

Figure 3: Ablation test of **KL-CPD**.

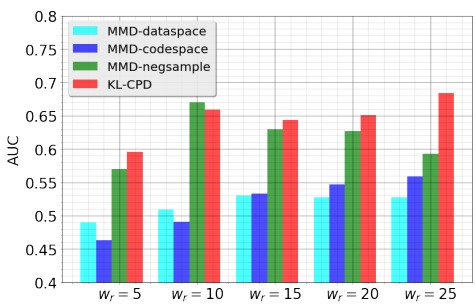

Figure 4: AUC vs. different window size $w_r$ on **Bee-Dance**.

The results are shown in Figure 3. We first notice the mild improvement of **MMD-codespace** over **MMD-dataspace**, showing that using MMD on the induced latent space is effective for discovering beneficial kernels for time series CPD. Next, we see **MMD-negsample** outperforms **MMD-codespace**, showing the advantages of injecting a random perturbation to the current interval to approximate $g_\theta(z|X^{(l)})$. This also justify the validity of the proposed lower bound approach by optimizing $M_k(\mathbb{P}, \mathbb{G})$, which is effective even if we adopt simple perturbed $\mathbb{P}$ as $\mathbb{G}$. Finally, **KL-CPD** models the $\mathbb{G}$ with an auxiliary generator $g_\theta$ to obtain conditional samples that are more complex and subtle than the perturbed samples in **MMD-negsample**, resulting in even better performance.

In Figure 4, we also demonstrate how the tolerance of delay $w_r$ influences the performance. Due to space limit, results other than **Bee-Dance** dataset are omitted, given they share similar trends. **KL-CPD** shows competitive AUC mostly, only slightly decreases when $w_r = 5$. **MMD-dataspace** and **MMD-codespace**, in contrast, AUC degradation is much severe under low tolerance of delay ($w_r = \{5, 10\}$). The conditional generated samples from **KL-CPD** can be found in Appendix B.5.

## 6 IN-DEPTH ANALYSIS ON SIMULATED DATA

To further explore the performance of **KL-CPD** with controlled experiments, we follow other time series CPD papers (Takeuchi & Yamanishi, 2006; Liu et al., 2013; Matteson & James, 2014) to create three simulated datasets each with a representative change-point characteristic: jumping mean, scaling variance, and alternating between two mixtures of Gaussian (**Gaussian-Mixtures**). More description of the generated process see Appendix B.2.

| Method | Jumping-Mean | Scaling-Variance | Gaussian-Mixtures |
|---|---|---|---|
| **ARMA** | 0.7731 (0.06) | 0.4801 (0.07) | 0.5035 (0.08) |
| **ARGP** | 0.4770 (0.03) | 0.4910 (0.07) | 0.5027 (0.08) |
| **RNN** | 0.5053 (0.03) | 0.5177 (0.08) | 0.5053 (0.08) |
| **LSTNet** | 0.7694 (0.09) | 0.4906 (0.07) | 0.4985 (0.07) |
| **ARGP-BOCPD** | 0.7983 (0.06) | 0.4767 (0.08) | 0.5027 (0.08) |
| **RDR-KCPD** | 0.6484 (0.11) | 0.7574 (0.06) | 0.6022 (0.11) |
| **Mstats-KCPD** | 0.7309 (0.05) | 0.7534 (0.04) | 0.6026 (0.08) |
| **KL-CPD** | **0.9454** (0.02) | **0.8823** (0.03) | **0.6782** (0.05) |

Table 3: AUC on three artificial datasets. Mean and standard deviation under 10 random seeds.

### 6.1 MAIN RESULTS ON SIMULATED DATA

The results are summarized in Table 3. **KL-CPD** achieves the best in all cases. Interestingly, retrospective-CPD (**ARGP-BOCPD**, **RDR-KCPD**, **Mstats-KCPD**) have better results compared to real-time CPD (**ARMA**, **ARGP**, **RNN,LSTNet**), which is not the case in real-world datasets. This suggests low reconstruction error does not necessarily lead to good CPD accuracies.

As for why **Mstats-KCPD** does not have comparable performance as **KL-CPD**, given that both of them use MMD as distribution distance? Notice that **Mstats-KCPD** assumes the reference time series (training data) follows the same distribution as the current interval. However, if the reference time series is highly non-stationary, it is more accurate to compute the distribution distance between the latest past window and the current window, which is the essence of **KL-CPD**.

### 6.2 MMD VERSUS DIMENSIONALITY OF DATA

We study how different encoders $f_\phi$ would affect the power of MMD versus the dimensionality of data. We generate an simulated time series dataset by sampling between two multivariate Gaussian $\mathcal{N}(0, \sigma_1^2 I_d)$ and $\mathcal{N}(0, \sigma_2^2 I_d)$ where the dimension $d = \{2, 4, 6, \ldots, 20\}$ and $\sigma_1 = 0.75, \sigma_2 = 1.25$.

Figure 5 plots the one-dimension data and AUC results. We see that all methods remain equally strong in low dimensions ($d \leq 10$), while **MMD-dataspace** decreases significantly as data dimensionality increases ($d \geq 12$). An explanation is non-parametric statistical models require the sample size to grow exponentially with the dimensionality of data, which limits the performance of **MMD-dataspace** because of the fixed sample size. On the other hand, **MMD-codespace** and **KL-CPD** are conducting kernel two-sample test on a learned low dimension codespace, which moderately alleviates this issue. Also, **KL-CPD** finds a better kernel (embedding) than **MMD-codespace** by optimizing the lower bound of the test power.

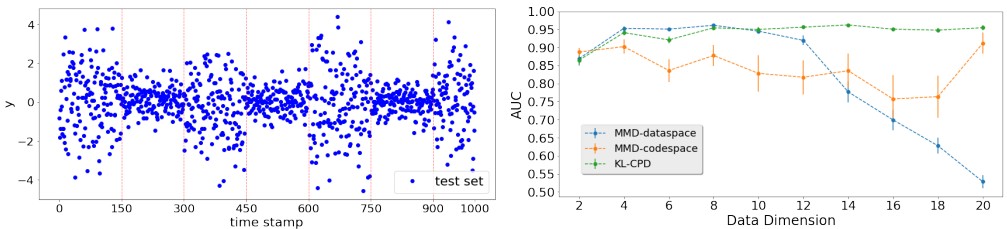

Figure 5: MMD with different encoder $f_{\phi_e}$ versus data dimension, under 10 random seeds.

## 7 CONCLUSION

We propose **KL-CPD**, a new kernel learning framework for two-sample test by optimizing a lower bound of test power with a auxiliary generator, to resolve the issue of insufficient samples in change-points detection. The deep kernel parametrization of **KL-CPD** combines the latent space of RNNs with RBF kernels that effectively detect a variety of change-points from different real-world applications. Extensive evaluation of our new approach along with strong baseline methods on benchmark datasets shows the outstanding performance of the proposed method in retrospective CPD. With simulation analysis in addition we can see that the new method not only boosts the kernel power but also evades the performance degradation as data dimensionality increases.

## 8 ACKOWLEDGEMENT

We thank the reviewers for their helpful comments. This work is supported in part by the National Science Foundation (NSF) under grant IIS-1546329, and by the United States Department of Energy via the Brookhaven National Laboratory under Contract No. 322174.

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

## A   CONNECTION TO MMD GAN

Although our proposed method **KL-CPD** has a similar objective function as appeared in **MMD GAN** (Li et al., 2017), we would like to point out the underlying interpretation and motivations are radically different, as summarized below.

The first difference is the interpretation of inner maximization problem $\max_k M_k(\mathbb{P}, \mathbb{G})$. MMD GANs (Li et al., 2017; Bińkowski et al., 2018) treat whole maximization problem $\max_k M_k(\mathbb{P}, \mathbb{G})$ as a new probabilistic distance, which can also be viewed as an extension of integral probability metric (IPM). The properties of the *distance* is also studied in Li et al. (2017); Arbel et al. (2018). A follow-up work (Arbel et al., 2018) by combining Mroueh et al. (2018) push $\max_k M_k(\mathbb{P}, \mathbb{G})$ further to be a scaled distance with gradient norm. However, the maximization problem (4) of this paper defines the lower bound of the test power, which also takes the variance of the empirical estimate into account, instead of the distance.

Regarding the goals, **MMD GAN** aims to learn a generative model that approximates the underlying data distribution $\mathbb{P}$ of interests. All the works (Dziugaite et al., 2015; Li et al., 2015b; Sutherland et al., 2017; Li et al., 2017; Bińkowski et al., 2018; Arbel et al., 2018) use MMD or $\max_k M_k(\mathbb{P}, \mathbb{G})$ to define distance, then try to optimize $\mathbb{G}$ to be as closed to $\mathbb{P}$ as possible. However, that is not the goal of this paper, where $\mathbb{G}$ is just an *auxiliary generative model* which needs to satisfies Eq. (3). Instead, we aim to find the most powerful $k$ for conducting hypothesis test. In practice, we still optimize $\mathbb{G}$ toward $\mathbb{P}$ because we usually have no prior knowledge (sufficient samples) about $\mathbb{Q}$, and we want to ensure the lower bound still hold for many possible $\mathbb{Q}$ (e.g. $\mathbb{Q}$ can be also similar to $\mathbb{P}$). However, even with this reason, we still adopt early stopping to prevent the auxiliary $\mathbb{G}$ from being exactly the same as $\mathbb{P}$.

## B   DETAILED EXPERIMENT SETTINGS

### B.1   BENCHMARK DATASETS

- **Bee-Dance**[2] records the pixel locations in x and y dimensions and angle differences of bee movements. Ethologists are interested in the three-stages bee waggle dance and aim at identifying the change point from one stage to another, where different stages serve as the communication with other honey bees about the location of pollen and water.

- **Fishkiller**[3] records water level from a dam in Canada. When the dam not functions normally, the water level oscillates quickly in a particular pattern, causing trouble for the fish. The beginning and end of every water oscillation (fish kills) are treated as change points.

- **HASC**[4] is a subset of the Human Activity Sensing Consortium (HASC) challenge 2011 dataset, which provides human activity information collected by portable three-axis accelerometers. The task of change point detection is to segment the time series data according to the 6 behaviors: stay, walk, jog, skip, stair up, and stair down.

- **Yahoo**[5] contains time series representing the metrics of various Yahoo services (e.g. CPU utilization, memory, network traffic, etc) with manually labeled anomalies. We select 15 out of 68 representative time series sequences after removing some sequences with duplicate patterns in anomalies.

### B.2   SYNTHETIC DATASETS

- **Jumping-Mean**: Consider the 1-dimensional auto-regressive model to generate 5000 samples $y(t) = 0.6y(t-1) - 0.5y(t-2) + \epsilon_t$, where $y(1) = y(2) = 0$, $\epsilon_t \sim \mathcal{N}(\mu, 1.5)$ is a Gaussian noise with mean $\mu$ and standard deviation 1.5. A change point is inserted at every

---

[2] http://www.cc.gatech.edu/~borg/ijcv_psslds/
[3] http://mldata.org/repository/data/viewslug/fish_killer/
[4] http://hasc.jp/hc2011
[5] https://webscope.sandbox.yahoo.com/catalog.php?datatype=s

$100 + \tau$ time stamps by setting the noise mean $\mu$ at time $t$ as

$$\mu_n = \begin{cases} 0 & n = 1, \\ \mu_{n-1} + \frac{n}{16} & n = 2, \ldots, 49, \end{cases}$$

where $\tau \sim \mathcal{N}(0, 10)$ and $n$ is a natural number such that $100(n-1) + 1 \leq t \leq 100n$.

- **Scaling-Variance**: Same auto-regressive generative model as **Jumping-Mean**, but a change point is inserted at every $100 + \tau$ time stamps by setting the noise standard deviation of $\epsilon_t$ at time $t$ as

$$\sigma_n = \begin{cases} 1 & n = 1, 3, \ldots, 49, \\ \ln(e + \frac{n}{4}) & n = 2, 4, \ldots, 48, \end{cases}$$

where $\tau \sim \mathcal{N}(0, 10)$ and $n$ is a natural number such that $100(n-1) + 1 \leq t \leq 100n$.

- **Gaussian-Mixtures**: Time series data are sampled alternatively between two mixtures of Gaussian $0.5\mathcal{N}(-1, 0.5^2) + 0.5\mathcal{N}(1, 0.5^2)$ and $0.8\mathcal{N}(-1, 1.0^2) + 0.2\mathcal{N}(1, 0.1^2)$ for every 100 time stamps, which is defined as the change points.

### B.3 COMPARING BASELINES

We include the following representative baselines in the literature of time series forecasting and change-point detection for evaluations:

- Autoregressive Moving Average (**ARMA**) (Box, 2013) is the classic statistical model that predicts the future time series based on an Autoregressive (AR) and a moving average (MA), where AR involves linear regression, while MA models the error term as a linear combination of errors in the past.

- Autoregressive Gaussian Process (**ARGP**) (Candela et al., 2003) is a Gaussian Process for time series forecasting. In an ARGP of order $p$, $x_{t-p:t-1}$ are taken as the GP input while the output is $x_t$. ARGP can be viewed as a non-linear version of AR model.

- Recurrent Neural Networks (**RNN**) (Cho et al., 2014) are powerful neural networks for learning non-linear temporal dynamical systems. We consider gated recurrent units (GRU) in our implementation.

- **LSTNet** (Lai et al., 2018) is a recent state-of-the-art deep neural network fore time series forecasting. LSTNet combines different architectures including CNN, RNN, residual networks, and highway networks.

- **ARGP-BOCPD** (Saatçi et al., 2010) is an extension of the Bayesian online change point detection (BOCPD) which uses ARGP instead of AR in underlying predictive models of BOCPD framework.

- **RDR-KCPD** (Liu et al., 2013) considers f-divergence as the dissimilarity measure. The f-divergence is estimated by relative density ratio technique, which involves solving an unconstrained least-squares importance fitting problem.

- **Mstats-KCPD** (Li et al., 2015a) consider kernel maximum mean discrepancy (MMD) on *data space* as dissimilarity measure. Specifically, It samples $B$ block of segments from the past time series, and computes $B$ times MMD distance between the past block with the current segment and takes the average as the dissimilarity measure.

### B.4 HYPERPARAMETER SETTINGS

For hyper-parameter tuning in **ARMA**, the time lag $p, q$ are chosen from $\{1, 2, 3, 4, 5\}$. For **ARGP** and **ARGP-BOCPD** the time lag order $p$ is set to the same as **ARMA** and the hyperparameter of kernel is learned by maximizing the marginalized likelihood. For **RDR-KCPD**, the window size $w$ are chosen from $\{25, 50\}$, sub-dim $k = 5$, $\alpha = \{0.01, 0.1, 1\}$. For **Mstats-KCPD** and **KL-CPD**, the window size $w = 25$, and we use RBF kernel with median heuristic setting the kernel bandwidth. The hidden dimension of GRU is $d_h = 10$ for **MMD-codespace**, **MMD-negsample** and **KL-CPD**. For **KL-CPD**, $\lambda$ is chosen from $\{0.1, 1, 10\}$ and $\beta$ is chosen from $\{10^{-3}, 10^{-1}, 1, 10\}$.

Our algorithms are implemented in Python (PyTorch Paszke et al. (2017)), and running on Nvidia GeForce GTX 1080 Ti GPUs. Datasets and experiment code are publicly available. For all the baseline methods we used the released source code, include MATLAB code[6] for **ARMA**, **ARGP** and **ARGP-BOCPD**, Pytorch code[7] for **RNN** and **LSTNet**, MATLAB code[8] for **RDR-KCPD**, and MATLAB code [9] for **Mstats-KCPD**.

## B.5 CONDITIONAL SAMPLES OF **KL-CPD**

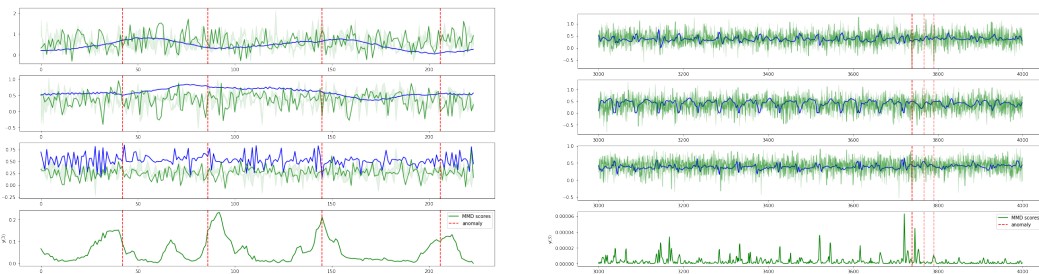

Figure 6: Conditionally generated samples by **KL-CPD** and system-predicted CPD scores on **Bee-Dance** (Left) and **HASC** (Right) datasets. In the first three subplots are ground truth signals (blue line), 10 conditional generated samples (green lines) and change points (red vertical line). The last subplot is MMD scores, which peaks around ground truth change points mostly.

---

[6]https://sites.google.com/site/wwwturnercomputingcom/software
[7]https://github.com/laiguokun/time_series_forecasting
[8]http://allmodelsarewrong.org/code/change_detection.zip
[9]https://www2.isye.gatech.edu/~yxie77/M_statistic_code.zip

