# OpenReview forum: "Kernel Change-point Detection with Auxiliary Deep Generative Models"
_ICLR.cc/2019/Conference_

### Official Review · AnonReviewer1 · 2018-10-23
**Not convinced that improvements are from better power**

**Rating:** 7
**Confidence:** 4

**Review:**

The manuscript entitled "Kernel Change-Point Detection with Auxiliary Deep Generative Models" describes a novel approach to optimising the choice of kernel towards increased testing power in this challenging machine learning problem.  The proposed method is shown to offer improvements over alternatives on a set of real data problems and the minimax objective identified is well motivated, however, I am not entirely convinced that (a) the performance improvements arise for the hypothesised reasons, and (b) that the test setting is of wide applicability.

A fundamental distinction between parametric and non-parametric tests for CPD in timeseries data is that the adoption of parametric assumptions allows for an easier introduction of strict but meaningful relationships in the temporal structure---e.g. a first order autoregressive model introduces a simple Markov structure---whereas non-parametric kernel tests typically imagine samples to be iid (before and after the change-point).  For this reason, the non-parametric tests may lack robustness to certain realistic types of temporal distributional changes: e.g. in the parameter of an autoregressive timeseries.  On the other hand, it may be prohibitively difficult to design parametric models to well characterise high dimensional data, whereas non-parametric models can typically do well in high dimension when the available data volumes are large.  In the present application it seems that the setting imagined is for low dimensional data of limited size in which there is likely to be non-iid temporal structure (i.e., outside the easy relative advantage of non-parametric methods).  For this reason it seems to me the key advantage offered by the proposed approach with its use of a distributional autoregressive process for the surrogate model may well be to introduce robustness against Type 1 errors due to otherwise unrepresented temporal structure in the base distribution (P).  In summarising the performance results by AUC it is unclear whether it is indeed the desired improvement in test power that offers the advantages or whether it is in fact a decrease in Type 1 errors.

Another side of my concern here is that I disagree with the statement: "As no prior knowledge of Q ... intuitiviely, we have to make G as close to P as possible" interpretted as a way to maximise test power; as a way to minimise Type 1 errors, yes.

Across change-point detection methods it is also important to distinguish key aspects of the problem formulation.  One particular specification here is that we have already some labelled instances of data known to come from the P distribution, and perhaps also some fewer instances of data labelled from Q.  This is distinct from fully automated change point detection methods for time series such as automatic scene selection in video data.  Another dissimilarity to that archetypal scenario is that here we suppose the P and Q distributions may have subtle differences that we're interested in; and it would also seem that we assume there is only one change-point to detect.  Or at least the algorithm does not seem to be designed to be applied in a recursive sense as it would be for scene selection.

Finally there is no discussion here of computational complexity and cost?

---

> ### Author Response · Authors · 2018-11-09
> **RE: Not convinced that improvements are from better power**
>
> Thank you very much for your valuable comments. We would try to address your concerns as below.
>
> We agree with you that there are multiple settings for change point detection (CPD) where samples could be piecewise iid, non-iid autoregressive, and more. It is truly difficult to come up with a generic framework to tackle all these different settings. In this paper, following the previous CPD works [1,2,3,4], we stay with the piecewise iid assumption of the time series samples. Extending the current model to other settings, such as the scene detection task, is interesting and we leave it for future work.
>
> For the piecewise iid case, as shown in our toy experiment in Sec. 3, optimizing kernel using surrogate distribution G indeed leads to better test power when samples from Q are insufficient. This demonstrates the effectiveness of our kernel selection objective without any autoregressive/RNN modeling to control the Type-I error. In Table 3, It is even interesting to see that for the synthetic Jumping-Mean and Scaling-Variance datasets that are generated from an autoregressive model with non-iid temporal samples, the non-parametric methods (RDR-KCPD and Mstats-KCPD) without RNN modeling are also comparable, sometimes better, compared to the AR-based methods.
>
> For the non-iid temporal structure in real-world applications, the concern is the improvement coming from adopting RNN and controlling type-I error for model selection (kernel selection). Indeed, using RNN parameterized kernels (trained by minimizing reconstruction loss) buy us some gain compared to directly conduct kernel two-sample test on the original time series samples (Fig 3 cyan bar rises to blue bar), but we still have to do model selection to decide the parameters of RNN.  In Table 2, we studied a kernel learning baseline, OPT-MMD, that optimizing an RNN parameterized kernel by controlling type-I error but without the surrogate distribution. OPT-MMD is inferior to the KL-CPD that introduce the surrogate distribution with an auxiliary generator. On the other hand, from Table 2, we can also observe KL-CPD is better than other RNN alternatives, such as LSTNet.  Those performance gaps between KL-CPD, OPT-MMD (regularizing type-I only) and other RNN works indicate the proposed maximizing testing power framework via an auxiliary distribution serves as a good surrogate for kernel (model) selection.
>
> In summary, we agree with you part of the improvement coming from introducing RNN,  and that our framework does have some limitation for different CPD settings. For the choice of real-world applications, we mainly follow the CPD literature [2,4], which also made piecewise iid assumptions on the time series samples while applying their framework on the likely non-iid real-world datasets, but we still observe the improvement brought by KL-CPD. It would be interesting to develop a theoretical framework for kernel learning to deal with non-iid data, and we leave it as future work.
>
> [1] Kernel change-point analysis, NIPS 2009
> [2] Change-point detection in time-series data by relative density-ratio estimation, Neural Networks 2013
> [3] A nonparametric approach for multiple change point analysis of multivariate data, JASA 2014
> [4] M-statistic for kernel change-point detection, NIPS 2015

---

> > ### Comment · AnonReviewer1 · 2018-11-17
> > **addressed some concerns**
> >
> > The authors have made a detailed reply to my comments and addressed a number of my concerns.

---

### Official Review · AnonReviewer3 · 2018-11-05
**A interesting study of how to optimize kernel change-point detection algorithm.**

**Rating:** 8
**Confidence:** 4

**Review:**

A new approach to choose a kernel to maximize the test power, for the kernel change-point detection. This provides an extension to the two-sample version of the problem (Gretton et al. 2012b, Sutherland et al. 2017). The difficulty is caused by that there is very limited samples from the abnormal distribution. The idea is based on choosing a surrogate distributions using generative model. The idea makes sense although there seems to be not much detail in how to choose the surrogate distribution. There is a mechanism to study the threshold. Real-data and simulation demonstrates the good performance. I think the idea is really interesting and I am impressed by the completeness of the work.

---

> ### Author Response · Authors · 2018-11-10
> **RE: A interesting study of how to optimize kernel change-point detection algorithm.**
>
> Thank you for your review and appreciation of our work. We will provide more details about how surrogate distributions are approximated by the generative models in our revision.

---

### Official Review · AnonReviewer2 · 2018-11-05
**A very neat idea with strong results**

**Rating:** 8
**Confidence:** 3

**Review:**

+ Using a generative model as the surrogate distribution for kernel two-sample test is novel
+ An important and new application of deep generative models
+ Strong experiments on synthetic and real-world time series data sets
+ Very clear writing and explanation of the idea

- reply sample segments from both directions (past and future) while in the practical setting, CPD is usually sequential and in one directional
- lack theoretical understanding of the limit of the neural-generator in the kernel two-sample test

---

> ### Author Response · Authors · 2018-11-10
> **RE: A very neat idea with strong results**
>
> Thank you for your review and appreciation of our work.
>
> - It is true that for some real-world applications, real-time CPD should be applied where only past samples are observed and anomaly alarm should be made immediately after observing a new sample. This paper, on the other hand, focuses on retrospective CPD where samples from both directions (past and future) are available for anomaly detection. While this setting is not real-time, but it typically offers more robust anomaly detection.

---

### Public Comment · (anonymous) · 2018-11-28
**Interesting work, with a novel idea and strong experiment results**

This is a good work, with a novel idea and strong experiment results. Besides kernel selection, using RNN is also very interesting to me; in fact, I hardly see a motivation when samples are iid.

I have a few questions:

- problem setting: do you really test if a point is a change-point, or if the sequence may contain a change-point? A point which is not a change-point does not mean all the points are from P. It appears to me that the latter case is considered. If this is true, how about using the maximum partition strategy on $X^(r)$?

- the illustrative example in Sec 3: 1) in the insufficient sample case, do you also use the same number of samples from $P$ to compute the unbiased MMD estimate? With some experience on toy examples, the selected kernel in Sutherland, et al can still help when using the biased MMD. 2). In Sec. 3.2, the selected $G$ seems not consistent with the proposed method. In the proposed method, $G$ should be close but not too close to $P$, and is independent of $Q$. However, the selected $G$ depends on the parameter of the real $Q$.

- problem formulation: compared with the MMD GAN paper, the formulation in this paper has an additional form term $\hat M_k(X, X')$. So how does this term help in finding the kernel?

- test threshold approximation: can you give more details on this part?

Finally, look forward to a new version and the codes.

---

### Meta-Review · Area_Chair1 · 2018-12-16
**A good paper but short reviews**

**Confidence:** 2
**Recommendation:** Accept (Poster)

**Metareview:**

This paper proposes a new kernel learning framework for change point detection by using a generative model. The reviewers agree that the paper is interesting and useful for the community. One of the reviewer had some issues with the paper but those were resolved after the rebuttal. The other two reviewers have short reviews and somewhat low confidence, so it is difficult to tell how this paper stands among other that exist in the literature. Overall, given the consistent ratings from all the reviewers, I believe this paper can be accepted.